# Triple-Helical DNA in *Drosophila* Heterochromatin

**DOI:** 10.3390/cells7120227

**Published:** 2018-11-23

**Authors:** Eduardo Gorab

**Affiliations:** Department of Genetics and Evolutionary Biology, Institute of Biosciences, University of São Paulo, São Paulo 05508-090, Brazil; egorab@usp.br; Tel.: +7-055-011-30918061

**Keywords:** chromosomes, heterochromatin, *Drosophila*, triplex DNA, antibodies, thiazole orange, satellite repeats

## Abstract

Polynucleotide chains obeying Watson-Crick pairing are apt to form non-canonical complexes such as triple-helical nucleic acids. From early characterization in vitro, their occurrence in vivo has been strengthened by increasing evidence, although most remain circumstantial particularly for triplex DNA. Here, different approaches were employed to specify triple-stranded DNA sequences in the *Drosophila melanogaster* chromosomes. Antibodies to triplex nucleic acids, previously characterized, bind to centromeric regions of mitotic chromosomes and also to the polytene section 59E of mutant strains carrying the *brown dominant* allele, indicating that AAGAG tandem satellite repeats are triplex-forming sequences. The satellite probe hybridized to AAGAG-containing regions omitting chromosomal DNA denaturation, as expected, for the intra-molecular triplex DNA formation model in which single-stranded DNA coexists with triplexes. In addition, Thiazole Orange, previously described as capable of reproducing results obtained by antibodies to triple-helical DNA, binds to AAGAG repeats in situ thus validating both detection methods. Unusual phenotype and nuclear structure exhibited by *Drosophila* correlate with the non-canonical conformation of tandem satellite arrays. From the approaches that lead to the identification of triple-helical DNA in chromosomes, facilities particularly provided by Thiazole Orange use may broaden the investigation on the occurrence of triplex DNA in eukaryotic genomes.

## 1. Introduction

Triple-stranded nucleic acids are complexes formed by three DNA or RNA complementary chains and may also be composed of both RNA and DNA [1,2]. Triplexes have long been characterized in vitro [3] and evidence has been accumulating for the occurrence in vivo of these non-canonical nucleic acid structures. Computational searches have identified the widespread genomic distribution of DNA stretches displaying the potential to form three-stranded conformations [4]. Triplex DNA binding as well as unwinding proteins have been identified [5,6] and, in addition, several reports using antibodies to such helical structures have given support for their in vivo formation in eukaryotic genomes [7,8,9,10,11,12,13,14,15]. We have also provided insights into implications of triple-helices composed of DNA and RNA in the control of gene expression by combining immunocytochemical detection with molecular biology methods [16,17].

In addition to the characterization of proteins that probably act on triplex DNA in vivo, further evidence for triple-stranded DNA in chromosomes has been presented [18,19] but most data available in the literature remain still indirect since in vitro and in silico results, albeit important, do not prove complex formation in vivo. High resolution techniques such as NMR cannot be applied on chromatin or chromosomes. Regarding immunological detection of these nucleic acid configurations, usually chromosomal binding sites of anti-triplex antibodies do not specify which sequences are targeted. Recent efforts directed to the search of alternative tools for detecting triple-stranded nucleic acids led to the finding that Thiazole Orange (TO), previously characterized as a dye that bind to triplex DNA in gels, bound to chromosomal regions of dipterans previously shown to be labelled by antibodies to triplex nucleic acid structures [20]. However, consistent validation of the dye for triplex DNA detection has been prevented by the lack of information on the chromosome sequences interacting with TO.

Before trying to assess roles played by three-stranded DNA in chromosomes, reliable signatures from its formation need to be provided. In vitro as well as in silico data should be supported by additional results achieved by methods directed to the sequence identification. Tools that are able to characterize specific DNA sequences, which are in turn associated with structures and/or phenotypes, might be ideal in showing not only consistent data for such conformations but also to raise tangible hypotheses on triplex DNA functions. This scenario has not been constructed yet, particularly in dipteran biology. Here, triple-helical DNA formation in the genome of *Drosophila melanogaster* is suggested on the basis of results obtained by different methods that converge on specific sequences in heterochromatin. Among the techniques employed in this report, TO is introduced as a simple and reliable tool that facilitates triplex DNA detection in chromosomes. Tentative hypotheses on the functional involvement of three-stranded DNA structures in *Drosophila* heterochromatin are also presented.

## 2. Materials and Methods

### 2.1. Animals

*Drosophila melanogaster* Canton-S, *y*, w, *y*/*w* and *y/w/bw^D^* flies came from laboratory stocks.

### 2.2. Preparation of Chromosome Spreads

Salivary glands or brain ganglia previously incubated in hypotonic solution were fixed in ethanol-acetic acid (3:1) and squashed in 50% acetic acid. Alternatively, squashing was performed in TE containing proteinase K (Calbiochem, San Diego, CA, USA), 10 µg/mL), pH 7.0; digestion time was monitored under phase contrast microscopy. The slides were frozen in liquid nitrogen and stored in ethanol at −10 °C after coverslip removal. For RNase treatment, chromosome spreads were rehydrated in 1× TBS followed by incubation at room temperature with RNase A (Calbiochem, San Diego, CA, USA) diluted (0.2 mg/mL) in 2× SSC for 2 h. Additional enzymatic treatments were carried out at room temperature with a mixture of RNase A (Calbiochem, San Diego, CA, USA, 0.2 mg/mL) and RNase H (GE Healthcare, Chicago, IL, USA, 1 unit per slide) diluted in 1× PBS. In some experiments, RNase treatment was followed by digestion with proteinase K (Calbiochem, San Diego, CA, USA) diluted as above in 1× PBS. The slides were washed in 1× TBS prior to immunodetection or in situ hybridization.

### 2.3. Immunological Detection of Triple-Helical DNA

The slides were first left at room temperature in 1× TBS, 0.1% Triton X-100 (TBST), 2% low fat powdered milk for 20–30 min. Slide incubations with purified anti-poly(dA).poly(rU).poly(rU) [15] diluted 1:50 in TBST from a stock solution (0.5 mg/mL) were done in a moistened chamber at room temperature for 2 h. After washes in TBST, the slides were incubated for 1 h at room temperature with sheep IgG anti-rabbit IgG conjugated with rhodamine (Sigma Chemical Co., St. Louis, MO, USA) diluted1:100 in TBST solution. The slides were washed twice in TBST for 30 min, and finally in 1× TBS for 5 min. Chromosomes were stained with DAPI and the slides mounted in antifading medium (Vectashield, Vector Labs., Burlingame, CA, USA). An additional control for the antibody specificity was performed by adding either poly(dT).poly(dA).poly(dT), or poly(dT).poly(dA).poly(rU) or even poly(rU).poly(dA).poly(rU) complexes assembled as previously described (15) to the antibody dilutions (approximately 200 ng per slide). Such a procedure abolished fluorescence detection in chromosomes. Images were captured with an Axiovert II Photomicroscope equipped with a CCD camera (AxioFan MRm, Carl Zeiss, Oberkochen, Germany) and coupled to an image analysis software package (ISIS, Carl Zeiss, Oberkochen, Germany). For simultaneous triplex and in situ hybridization detection, image coordinates were registered for subsequent capture before washing slides for satellite probe hybridization and detection. Inspection was carried out in 12 slides, each containing polytene chromosome spreads from one larva (approximately 100 nuclei) and 8 slides each containing one brain ganglion (approximately 2000 cells). Mean values (90.1% ± 2.8%; 87.6% ± 2%) represent respectively the number of chromosomes or brain nuclei displaying labelling relative to all chromosomes or nuclei per slide. Data came from 4 polytene chromosome slides and 3 neuroblast slides taken by chance.

### 2.4. Thiazole Orange (TO) Staining

TO (Sigma-Aldrich Chemical Co., St. Louis, MO, USA ) stock solution (0.3 mM) was diluted in 1× PBS to a final concentration in the range of 20–60 nM following slide incubation at room temperature for 5 min and subsequent washing in 1× PBS for 20 min. The slides were mounted in anti-fading medium (Vectashield, Vector Labs, Burlingame, CA, USA) and inspected with epifluorescence optics (Nikon Instruments Inc., Melville, NY), using excitation wavelengths of 488 or 532 nm. Detection was carried out in 8 slides, each containing polytene chromosome spreads from one larva. Mean value of positive staining data (82.4% ± 2.4%) was obtained as described above from 3 slides taken at random. For simultaneous visualization of anti-triplex and TO staining, the slides were first processed for immunological detection as described above. Images were captured with an Axiophot 2 Photomicroscope equipped with a Zeiss CCD camera (AxioFan MRm, Carl Zeiss, Oberkochen, Germany) and coupled to an image analysis software package (ISIS, Zeiss, Carl Zeiss, Oberkochen, Germany). Chromosomal sites labelled by antibodies that were captured in the TRITC channel displayed no fluorescent signal in the FITC channel. Coordinates of each image were registered for subsequent inspection after TO staining. The coverslips were pried off by washing the slides in TBST for 1 h at room temperature and once in 1× TBS for 10 min. The chromosome spreads were then stained by TO as described above and the fluorescent signals captured in the FITC channel.

### 2.5. In Situ Hybridization

Satellite probe was synthesized by PCR using 1 µg of sat1a (5′AAGAGAAGAGAAGAG3′) and sat1b (5′CTCTTCTCTTCTCTT3′) primers in 50 µL solution containing enzyme, enzyme buffer, dATP, dCTP, dGTP and MgCl_2_ for a standard PCR reaction and 0.02 mM biotin-11-dUTP (Sigma Chemical Co., St. Louis, MO, US). Cycle conditions: 94 °C for 60 s, followed by 26 cycles of 93 °C for 60 s, 56 °C for 30 s, 70 °C for 20 s and a final extension at 72 °C for 60 s. PCR products as the result of extension of the asymmetrical dimmer formed after primer annealing were checked in agarose gels and were found in the range of 200–400 bp. Ribosomal gene (rDNA) probe of *Drosophila*, p*Dm*-238 [21], was labelled with digoxigenin-11-dUTP by random priming according to the instructions of the manufacturer (Roche, Rotkreuz, Switzerland). Labelling of the rDNA probe was successfully proven on chromosome spreads that underwent DNA denaturation. When chromosomal DNA denaturation step was not performed, the hybridization solution containing both, satellite repeats and rDNA probes was heat denatured for 5 min and then applied on the slides. The probe mixture consisted of labelling reaction product containing 40% formamide, 2× SSPE and 0.1% SDS to a final volume of 100 µL; 5–10 µL of the probe mixture was applied on each air-dried slide and covered with a plastic coverslip. The slides were steam heated at 75 °C for 5–10 min and immediately kept overnight in a closed box at 37 °C. The slides were then washed once in 0.1× SSPE, 0.2% SDS at 45 °C for 30 min, followed by incubation at room temperature in 1× TBS, 0.1% Triton X-100 (TBST) for 30 min. For fluorescent detection, either goat IgG anti-biotin TRITC-labelled antibody (Sigma) and/or FITC-labelled sheep IgG anti-digoxigenin (Roche, Rotkreuz, Switzerland) was diluted 1:100 in TBST/Superblock solution. Slide incubation was done in a moistened chamber at room temperature for 1 h. The slides were then washed twice in TBST for 10 min and finally in 1× TBS for 5 min. The slides were mounted in antifading medium (Vectashield, Vector Labs) and inspected with epifluorescence optics (Carl Zeiss, Oberkochen, Germany). Detection was carried out in 15 slides, each containing polytene chromosome spreads from one larva and 8 slides each containing one brain ganglion. Polytene slides (5) and brain slides (3) taken by chance were used as samples for counting hybridization signals relative to the total number of either polytene chromosomes (74.6% ± 3.1%) or brain nuclei (66.7% ± 2.5%). Mean values of positive labelling results were obtained as described above.

### 2.6. Triplex DNA Search Algorithm

“Triplex” is an “R/Bioconductor” software package [22] that allows identification and visualization of potential intra-molecular triplex patterns in DNA sequences.

## 3. Results

### 3.1. Triplex DNA is Immunologically Detected in Centromeric Regions of Drosophila Chromosomes

Triple-helical DNA formation in *Drosophila melanogaster* chromosomes was first assessed using antibodies raised to the poly-(rU).poly-(dA).poly-(rU) complex. This antibody was extensively characterized in relation to their ability to recognize different three-stranded configurations [15,17] (Figure 1) and indicated the occurrence of triplex DNA in pericentric heterochromatin of *D. melanogaster* polytene chromosomes [15].

As further controls, chromosome spreads underwent both, RNase and proteinase treatments prior to immunological detection of triple-helical DNA. Time course digestion by proteinase K was limited to ten minutes as longer enzyme action led to the loss of chromosome morphology. The results showed that the treatments performed did not affect triple-stranded DNA detection and were essentially the same observed when enzymatic treatments were omitted. This is an indication that neither RNA nor proteins in the chromosome structure are antibody targets (Figure 2).

The antibody labelling in giant chromosomes of this species is not spread over the entire heterochromatic region, suggesting that specific sequences were targeted. As DNA under-replication is present in polytene heterochromatin but does not seem to occur in diploid nuclei, the immunological detection was made using mitotic spreads from neuroblasts that could be informative despite the dimensions of mitotic chromosomes. Strong signals were seen in chromosome 2, regions h37-39 while faint fluorescence was detected in the region h57 in chromosome 3 (Figure 3). Antibody binding was also observed in chromosome Y, particularly in h5, h21, h23 and h25 regions (Appendix A).

### 3.2. Triplex-Forming Sequences are Identified by Exploiting Immunocytochemical Detection and Drosophila Mutants

The labelling pattern obtained with antibodies to triplex nucleic acids was similar to that observed for the location of AAGAG and AAGAGAG repeats in mitotic chromosomes of *Drosophila* (Appendix A) but simple data comparison cannot reveal the antibody target unambiguously. Sequences reactive with anti-triplex antibodies were identified in chromosomes from larvae carrying the *brown^Dominant^* (*bw^D^*) allele, characterized by the AAGAG/AAGAGAG tandem repeat block inserted into the polytene section 59E [23], which appeared strongly labelled by the antibodies (Figure 4, Appendix A). Section 59E is not labelled by anti-triplex antibodies in chromosomes from wild type larvae. In the latter strains, antibody signals are restricted to the pericentric heterochromatin (Figure 4a–c).

The mutant strain that was used in the experiments is in fact a triple mutant as it carries, in addition to *bw^D^*, *y* and *w* alleles. Controls performed with *y*, *w* and *y/w* mutants lacking the *bw^D^* allele displayed no signal at polytene section 59E, showing that the *bw^D^* mutation is required for triplex DNA detection in the *brown* locus (Figure 5).

### 3.3. Conceptual Intra-Molecular Triplex DNA Assembled with AAGAG Repeats

Assuming that intra-molecular triplexes [2] are formed in *Drosophila* heterochromatin, prediction in silico was performed by loading AAGAG tandem repeats into “Triplex” software package [22]. Algorithm choice was made on the grounds of its design specialized in identifying potential intra-molecular DNA triplexes. While puric and pyrimidic triple helices can conceptually be formed according to the software output, the pyrimidic triplex DNA was found to display significantly high scores (Figure 6, Appendix A).

### 3.4. A Structural Mark of Triple-Helical DNA is Detected by *In Situ* Hybridization

In the case of intra-molecular triplexes, the triplex forming strand is provided by one of the strands of the same duplex DNA and, consequently, single-stranded DNA is also generated. This structural feature was exploited as a diagnostic marker for intra-molecular triplex DNA formation since single-stranded DNA should hybridize to specific probes when chromosomal DNA denaturation is omitted. Satellite probe hybridization was readily detected either in salivary gland squashes (Figure 7) or in nuclei from brain ganglia (Appendix A) prepared at pH 7 without denaturing chromosomal DNA. Ribosomal DNA probe added as a control in the hybridization mixture provided no signal (Figure 8) unless chromosomal DNA is previously denaturized (Appendix A)

### 3.5. Thiazole Orange and Antibodies to Triplex DNA Simultaneously Bind to Satellite Repeats

Dyes have been used in biology of the nucleus since the nineteenth century and have since provided significant information on chromosome features. However, no dye has been characterized as capable of binding specifically to non-canonical nucleic acid structures such as triple helices in chromosomes. In a recent report, Thiazole Orange (TO) bound to the same chromosomal regions of dipterans previously found to be reactive with antibodies to triplex nucleic acid structures [20]. Nevertheless, the same limitation as observed with the antibody occurred with TO, namely the target sequences remained unknown so that the specificity of both methods has continued as an open question. Polytene chromosomes from wild type *Drosophila* larvae and those carrying the *bw^D^* allele were then exploited in order to test the TO potential to bind to three-stranded DNA. The results remarkably matched those obtained by immunocytochemistry as only pericentric heterochromatin of wild type larvae was preferentially stained by TO (Figure 9).

Also, polytene section 59E of *bw^D^* mutants was clearly reactive with the dye (Figure 10 and Figure 11). TO reproduced the same results obtained in mitotic chromosomes by immunocytochemistry, as shown in Figure 3 and Appendix A, although displaying less intense signals (data not shown).

## 4. Discussion

Centromeric as well as peri-centric regions of *Drosophila melanogaster* chromosomes are heterochromatic. In cells that undergo polyteny, these regions aggregate into a structure named chromocenter [24]. Compared to mitotic chromosomes, centromeric and peri-centric areas are under-represented in polytene chromosomes as a result of DNA under-replication that has long been described in *Drosophila* [25]. Therefore, triple-helical DNA detection as well as in situ hybridization results using DNA probes for the heterochromatin display different results in mitotic and polytene chromosomes. Moreover, as the chromocenter lacks banding pattern, in addition to DNA under-representation, in situ hybridization, anti-triplex DNA or even protein detection results are not seen as bands in this region. With particular regard to the *bw^D^* mutation, AAGAG tandem repeats are translocated from the heterochromatin to the euchromatic region of chromosome 2 containing *brown* [23]. This leads to poorer representation of these satellite sequences in heterochromatin, a feature significantly enhanced in salivary gland nuclei as well as in other polytene tissues as a consequence of DNA under-replication. For this reason, triple-helical DNA detection by any method employed in this work is considerably more difficult in heterochromatin of polytene chromosomes from *bw^D^* flies.

Previous observations on the potential of poly-puric/poly-pyrimidic DNA sequences to adopt triple-helical configurations [2,26] added to triplex algorithm data introduced in this work (Figure 6, Appendix A) reinforce the conclusions drawn in this report as the base composition of satellite repeats meets that requirement. Furthermore, experimental support for this assumption also came from DNA association assays in vitro demonstrating that AAGAGAG repeats form three-stranded structures [27,28]. These repeats are closely related to AAGAG satellites and both coexist in centromeric regions of *Drosophila* chromosomes [29]. Hence, data from distinct sources strengthen the helical features of centromeric satellites as presented here.

The non-canonical DNA conformation of the satellite arrays correlates with a specific phenotype as well as a nuclear structure of *Drosophila*, both unusual since they cannot be explained by comparison to related characteristics well studied in this organism. After translocation of the AAGAG repeat block into the section 59E, gene silencing of the resulting *bw^D^* allele is also spread *in trans* over the wild type allele in heterozygous flies [30]. It has been stated that the repressive environment of the heterochromatin is only exerted *in cis* and results from the interaction between Heterochromatin Protein 1 (HP1) and methylation of histone H3 at lysine 9 [31]. The hypothetic heterochromatic inactivation *in trans* would require H3 methylation that is lacking within the translocated repeat block [30] and it is known that HP1 alone does not lead to heterochromatin formation as it may appear enriched in transcriptionally active euchromatic sites [32]. Therefore, other components in the duo formed by the somatic pairing of the two alleles operate to keep *bw^+^ trans*-repressed. Although one cannot establish a cause-effect relationship at present, the helical structure of the AAGAG tandem repeat in the *bw^D^* allele is a putative candidate for silencing *bw^+^ in trans*, either alone or together with other molecular entities.

Also, unusual histone tetramers (“hemisomes”) in place of typical octamers have been characterized in centromeric heterochromatin of *Drosophila* [33]. As triple-stranded DNA excludes canonical nucleosomes [34,35], the possibility that AAGAG repeats take part in the “hemisome” building could be tested experimentally.

Different techniques leading to converging results reinforce diagnosis of triplex formation and, among them, use of TO as a triple-helical DNA probe was introduced as an additional tool to strengthen the data obtained with bioinformatics, antibodies and in situ hybridization. Moreover, its potential to broaden the screening of triplex DNA in chromosomes is high as TO is extremely easy and rapid to be used. Given the widespread occurrence of homopuric/homopyrimidic DNA stretches in eukaryotic nuclei, triplex DNA configuration is likely to be more frequent than expected.

## Figures and Tables

**Figure 1 cells-07-00227-f001:**
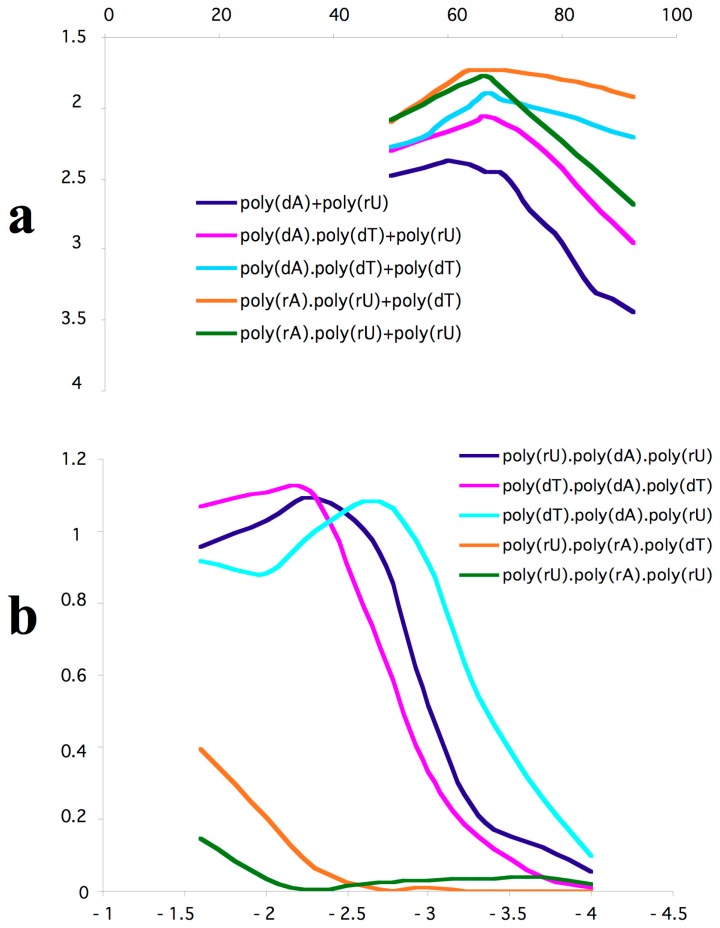
Mixing curves of homopolymeric nucleic acid complexes assembled by adding either poly-deoxyadenylic acid or poly-uridilic acid (**a**). Cumulative percentage of poly-pyrimidic nucleic acid addition (X axis) was plotted relative to the corresponding absorbance measured at 260 nm (Y axis). The reactivity of samples that displayed the highest hypochromicity to antibodies to poly(rU).poly(dA).poly(rU) was measured in immunosorbent assays (**b**). Curves were obtained by plotting logarithmic values of antibody concentrations (X axis) and the corresponding absorbance measured at 405 nm (Y axis). Data for graph construction were taken from [15].

**Figure 2 cells-07-00227-f002:**
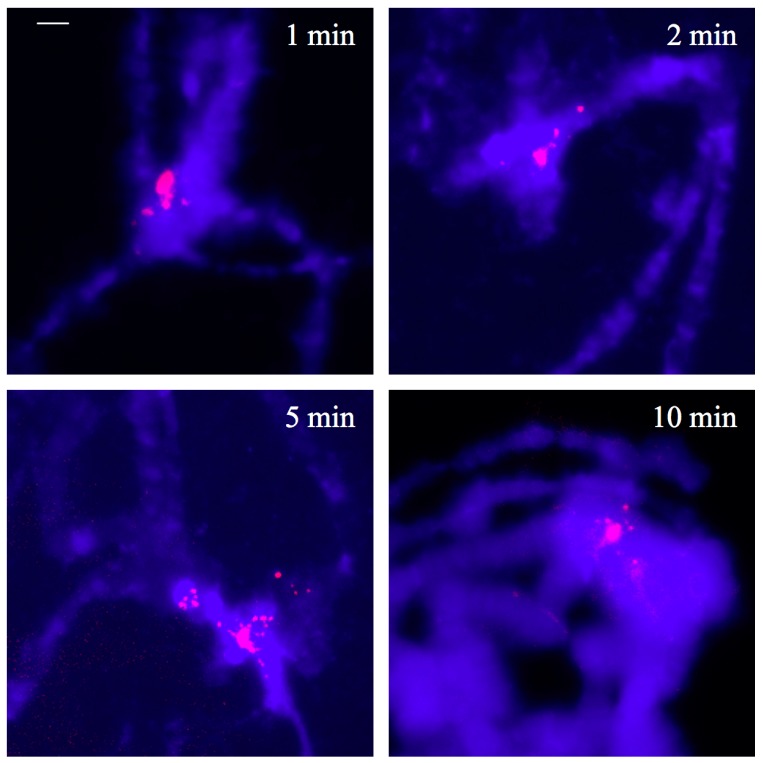
Polytene chromosome spreads of *D. melanogaster* wild type were treated with RNase A/RNase H mixture followed by proteinase K digestion in a time course experiment and subsequent immunological detection of triple-stranded DNA. DAPI staining (blue signal) and antibody labelling (red signal) were superimposed. Scale bar represents 25 µm.

**Figure 3 cells-07-00227-f003:**
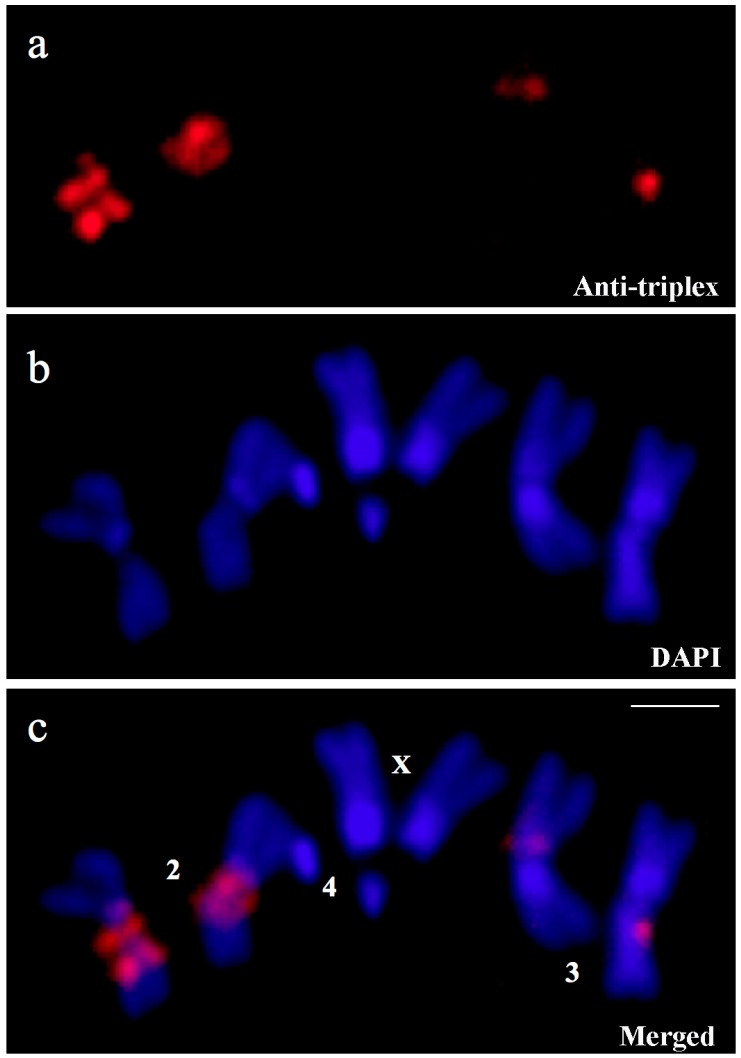
Neuroblast chromosomes from wild type larva stained with antibodies (**a**, Anti-triplex), chromosomal DNA staining with DAPI (**b**, DAPI) and the corresponding merged signals (**c**, Merged) showing centromeric regions of *D. melanogaster* chromosomes reactive to anti-triplex DNA antibodies. Chromosomes appear identified (2, 3, 4, X). Scale bar corresponds to 5 μm.

**Figure 4 cells-07-00227-f004:**
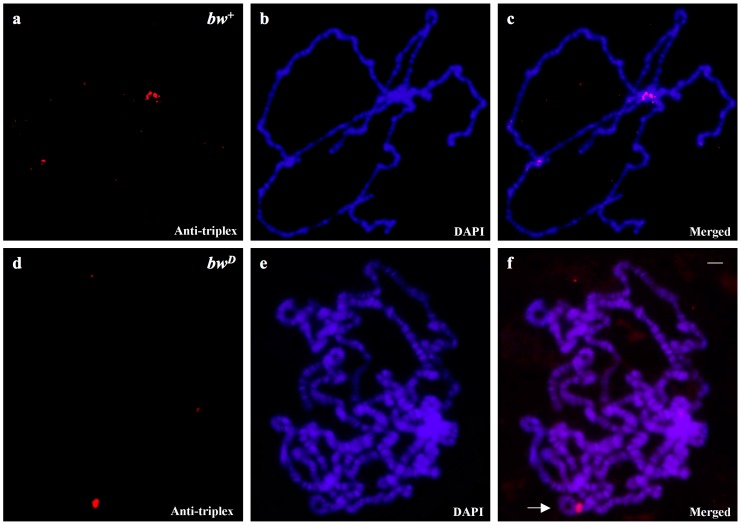
Polytene chromosomes of *D. melanogaster* from wild type (*bw*^+^) and *y/w/bw^D^* (*bw^D^*) larvae are shown. Antibody staining (**a**,**d**, Anti-triplex), DAPI staining (**b**,**e**, DAPI) and the corresponding merged signals (**c**,**f**, Merged). The arrow points to the section 59E carrying the bw*^D^* allele that is strongly reactive to anti-triplex antibodies. Scale bar corresponds to 20 μm.

**Figure 5 cells-07-00227-f005:**
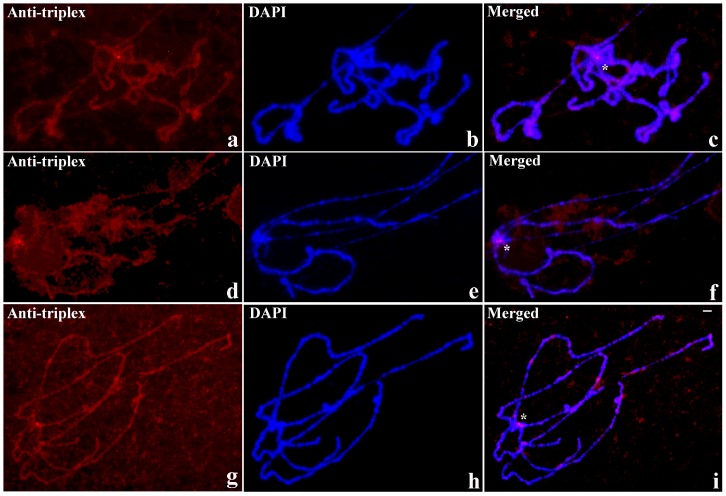
Anti-triplex DNA detection in polytene chromosomes of *D. melanogaster* from *y* (**a**–**c**), *w* (**d**–**f**) and *y/w* (**g**–**i**) mutants. Antibody labelling (**a**,**d**,**g**, Anti-triplex), DAPI staining (**b**,**e**,**h**, DAPI) and the corresponding merged signals (**c**,**f**,**i**, Merged). Anti-triplex signals are restricted to the pericentric heterochromatin (close to the asterisks). Scale bar corresponds to 25 μm.

**Figure 6 cells-07-00227-f006:**
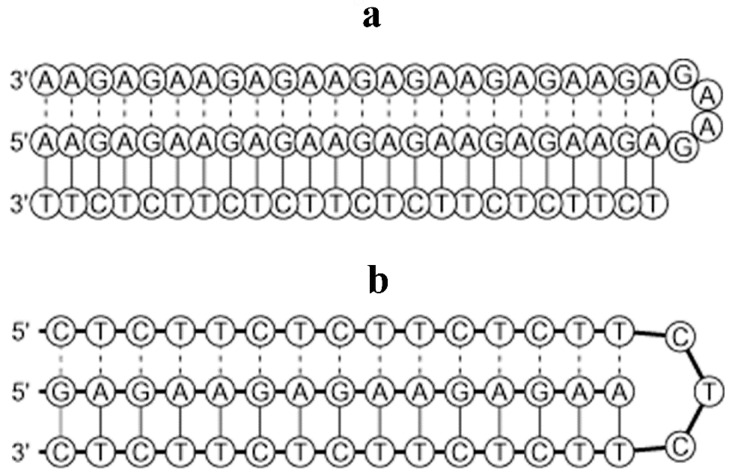
Conceptual intra-molecular triple helices formed by AAGAG tandem repeats. 2D diagrams were obtained after loading AAGAG tandem repeats into the “Triplex” software package [22]; purine triplex (**a**); pyrimidine triplex (**b**). Additional details on the program output in Appendix A.

**Figure 7 cells-07-00227-f007:**
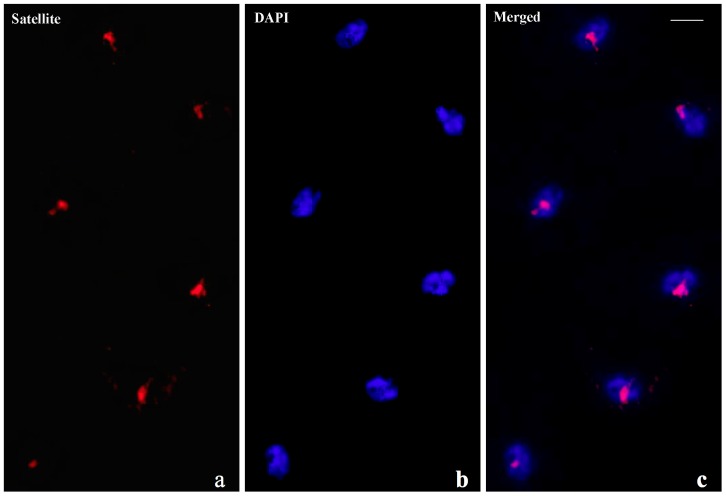
AAGAG repeat probe hybridization to salivary gland nuclei of *D. melanogaster* carrying the *bw^D^* allele prepared in neutral pH omitting chromosomal DNA denaturation. Hybridization detection (**a**, Satellite), chromosomal DNA staining (**b**, DAPI) and the corresponding merged signals (**c**, Merged).

**Figure 8 cells-07-00227-f008:**
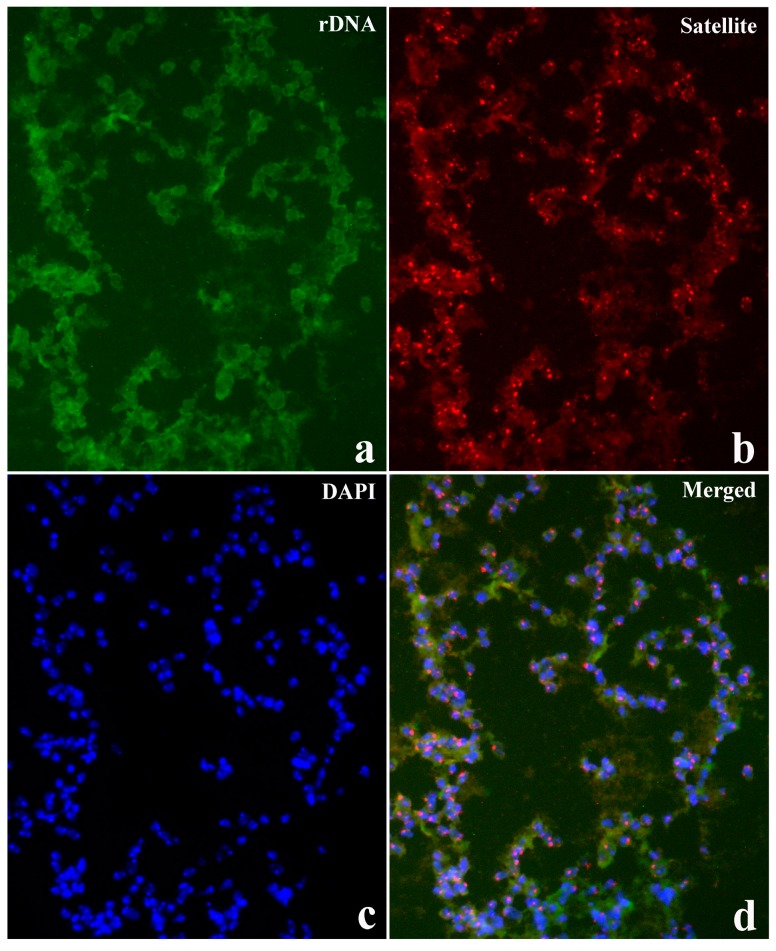
Simultaneous ribosomal DNA and AAGAG repeat probe hybridization to *Drosophila* neuroblast nuclei carrying the *bw^D^* allele prepared in pH 7.0 omitting chromosomal DNA denaturation. Ribosomal DNA probe channel (**a**, rDNA), AAGAG probe channel (**b**, Satellite), nuclei stained with DAPI (**c**, DAPI) and the corresponding merged signals (**d**, Merged).

**Figure 9 cells-07-00227-f009:**
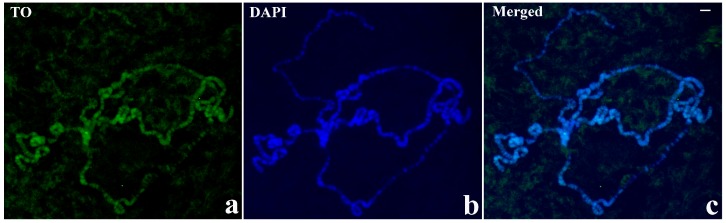
TO and DAPI staining in polytene chromosomes of *D. melanogaster* wild type. TO staining (**a**, TO). The corresponding DAPI staining (**b**, DAPI) and the two superimposed images(**c**, Merged). Scale bar corresponds to 25 μm.

**Figure 10 cells-07-00227-f010:**
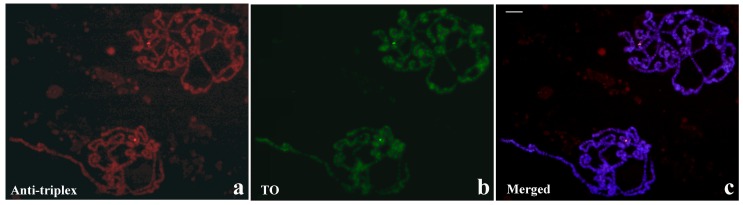
Simultaneous detection of anti-triplex and TO staining in polytene chromosomes from *y/w/bw^D^* (*bw^D^*) larva. Antibody labelling (**a**, Anti-triplex), TO staining (**b**, TO) and merged signals of antibody, TO and DAPI staining (**c**, Merged). The brightest signals fall in a single section of the polytene complement showing the overlap by TO and immunocytochemical labelling. Scale bar corresponds to 40 μm.

**Figure 11 cells-07-00227-f011:**
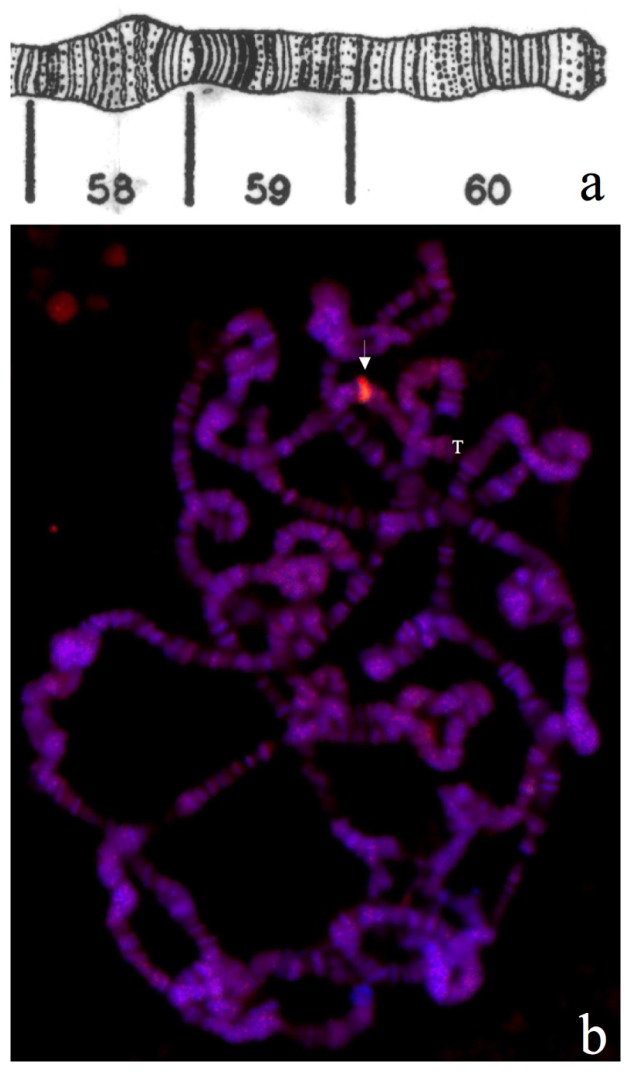
Drawing of chromosome sections 58-60 in arm 2R modified from [24]. (**a**). Magnification of one polytene complement from the top of Figure 10 displaying merged signals of antibody, TO and DAPI staining (arrow) (**b**) showing in detail that polytene chromosome section 59E carrying the *bw^D^* allele is reactive to TO. Arm 2R telomere at chromosome section 60 is indicated (T).

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
