# Peer review of "Triple-Helical DNA in Drosophila Heterochromatin"

_cells, 2018, doi:10.3390/cells7120227_

Reviewer 1 Report

As I have explained previously, demonstrating the specificity of the antibodies is a central and necessary part of the immunohistological staining procedure. The author now added such a control to the method section. To make the study convincing the control has to been shown as part of at least one figure (e. g. Fig 2 or Fig. 3). 

From the description of the control I understand that the triple helical DNAin excess is added first to the antibody solution. Secondly the mixture of antibodies and triple helical DNA is incubated with the samples. To be meaningful triple helical DNA is added first to the sample, secondly the antibody is added to the sample preexposed to the triple helical DNA. The reason why the order is important is that the antibodies should have the choice between competitor triple helical DNA and the sample. Adding the triple helical DNA before the sample to the antibody does not allow for a competition.

Author Response

As I have explained previously, demonstrating the specificity of the antibodies is a central and necessary part of the immunohistological staining procedure. The author now added such a control to the method section. To make the study convincing the control has to been shown as part of at least one figure (e. g. Fig 2 or Fig. 3). 

From the description of the control I understand that the triple helical DNAin excess is added first to the antibody solution. Secondly the mixture of antibodies and triple helical DNA is incubated with the samples. To be meaningful triple helical DNA is added first to the sample, secondly the antibody is added to the sample preexposed to the triple helical DNA. The reason why the order is important is that the antibodies should have the choice between competitor triple helical DNA and the sample. Adding the triple helical DNA before the sample to the antibody does not allow for a competition.

 R- Regarding controls, two new figures (1,2) were added according to your request. We have never tried the experimental approach you have described above namely the study of possible interactions between external triplex conformations and chromosome structure as far as it is preserved by usual fixatives. The rationale for using triplex excess added to the antibody solution is correlated to what is shown in Figure 2 namely, the antibody target is not RNA or proteins. Labelling results after previously reacting antibodies with exogenous triplexes would raise doubts on the chromosomal target and consequently on the antibody specificity.  From the negative results observed, one may conclude that the triplex/antibody complex is unable to react with the chromosome structure but one cannot predict the behavior of triplexes alone exposed to the chromosome structure. I think this would be another study and hypothetical results would argue for exogenous triplexes bound to chromosomal DNA, RNA or even proteins and would not work as a control for antibody binding in that case.

Reviewer 2 Report

I found that the author has appropriately responded to all of my comments. The manuscript is nicely improved. I recommend this paper to be accepted after minor revision.

 Minor changes:

1. Line 186 ‘RNAse’ is supposed to be ‘RNase’.

2. To make it easy to understand for the readers, it is better to label each picture. For example, add labels of ‘triplex antibodies’, ‘DAPI’, and ‘merged’ to panels a, b, and c, respectively, in Figure 3.

Author Response

I found that the author has appropriately responded to all of my comments. The manuscript is nicely improved. I recommend this paper to be accepted after minor revision.

 Minor changes:

1.  Line 186 ‘RNAse’ is supposed to be ‘RNase’.

R- Thank you for pointing out the mistake, it is corrected in the version submitted.

1.  To make it easy to understand for the readers, it is better to label each picture. For example, add labels of ‘triplex antibodies’, ‘DAPI’, and ‘merged’ to panels a, b, and c, respectively, in Figure 3.

R- Additional labellings were made in Figures 3, 4, 5, 7, 8, 9, 10, S1, S2, S3, S5 and S6. Changes were also made in the corresponding figure captions.

Round  1

Reviewer 1 Report

Dr. Gorab has recently reported that Thiazole Orange (TO) detects triplex DNA in salivary gland chromosomes of D. melanogaster and R. Americana (Gorab and Pearson 2018 J. Histochem. Cytochem.). The submitted paper extends the previous study and suggests that both the antibodies against triplex DNA and TO detect AAGAG-repeats which are predicted to form triplex DNA, by using the wild type and the mutant strain that contains the brown dominant allele. The paper is well written, but it would be better to address the following concerns before its acceptance.

 Major points:

To make it clear the connection between AAGAG-repeats and triplex DNA in vivo, it is better to perform double staining and provide the direct evidence showing that AAGAG-repeats colocalize with the signal of the antibodies against triplex DNA and/or TO.

The statistical evaluation is missing in this paper. It is better to provide the sample number the author examined and the number of the samples/nuclei that show the signal.

The author argues that “the triplex…consequently, single-strand DNA is also generated. This structural feature was exploited as a diagnostic marker of H-DNA formation…” (lines 170-173). However, single-stranded DNA is also created when R-loops (RNA:DNA hybrids) are formed. It is better to address this point. The author may treat the sample with RNaseH to destroy R-loops before the AAGAG repeat probe hybridization.

 Minor points:

Sometimes the author wrote (Vectashield, Vector Labs) (lines 80 and 118) but sometimes (Vectashield, Vector Lab.) (line 87). It is better to write in the same way.

I feel “identical” is too strong. “similar” would be better (line 143).

It is better to explain what H-DNA stands for (Abstract and line 158).

“Ribosomal DNA probe added as a control….” in lines 176-177. It would be better to provide this control data including the results of rDNA with or without heat denaturation in Supplementary Figures.

The reference paper [19] is not appropriate in the context “In a recent report, Thiazole Orange (TO) bound to … [19]. (lines 187-188).

There are two [22]s in the list of the References (lines 309-312).

It is not easy to recognize the signal of antibodies and TO in Figure 5. It is better to make the Figure 5 brighter.

Author Response

Reviewer 1

 Major points:

To make it clear the connection between AAGAG-repeats and triplex DNA in vivo, it is better to perform double staining and provide the direct evidence showing that AAGAG-repeats colocalize with the signal of the antibodies against triplex DNA and/or TO.

R- I would like to thank very much for carefully reviewing this submission and for your comments that will surely improve the manuscript. A new supplementary figure addressing the issue (Fig.S3) was added in the new manuscript version.

The statistical evaluation is missing in this paper. It is better to provide the sample number the author examined and the number of the samples/nuclei that show the signal.

R-Sample number was provided for each sub-section in Materials and Methods. Statistics is not necessary since the results are seen in the majority of chromosomes per slide. This information now appears in Materials and Methods.

The author argues that “the triplex…consequently, single-strand DNA is also generated. This structural feature was exploited as a diagnostic marker of H-DNA formation…” (lines 170-173). However, single-stranded DNA is also created when R-loops (RNA:DNA hybrids) are formed. It is better to address this point. The author may treat the sample with RNaseH to destroy R-loops before the AAGAG repeat probe hybridization.

R-We have long performed DNA/RNA hybrid detection in chromosomes using specific antibodies to hybrids (Science,1970, 169:609-11; Mol Immunol, 1982, 19:413-20). RNaseA concentration usually employed in our experiments and mentioned in Materials and Methods also digests RNA in hybrid form as no antibody signal is seen in controls performed with RNaseA. A sentence addressing this point was added in Materials and Methods (Preparation of chromosome spreads).

 Minor points:

Sometimes the author wrote (Vectashield, Vector Labs) (lines 80 and 118) but sometimes (Vectashield, Vector Lab.) (line 87). It is better to write in the same way.

R-Corrected in the new manuscript version.

I feel “identical” is too strong. “similar” would be better (line 143).

R-Your suggestion was followed in the new manuscript version.

It is better to explain what H-DNA stands for (Abstract and line 158).

R-H-DNA was deleted from the manuscript as it is an intramolecular triplex DNA synonym.

“Ribosomal DNA probe added as a control….” in lines 176-177. It would be better to provide this control data including the results of rDNA with or without heat denaturation in Supplementary Figures.

R-A supplementary figure (S6) addressing this point was included in the manuscript.

The reference paper [19] is not appropriate in the context “In a recent report, Thiazole Orange (TO) bound to … [19]. (lines 187-188).

There are two [22]s in the list of the References (lines 309-312).

R-Thank you for pointing the two mistakes. They were corrected in the new manuscript version.

It is not easy to recognize the signal of antibodies and TO in Figure 5. It is better to make the Figure 5 brighter.

R-Figure 5 was modified according to your suggestion.

 Reviewer 2 Report

Triple helices are unusual DNA and RNA structures, which have been studied mostly in vitro. The occurrence and the function of these structures in vivo is unclear. To my knowledge no good experimental system has been established to investigate triple helix structures in a physiological context and during development. Dr. Gorab’s takes a histological approach to investigate whether triple helix structures can be detected in Drosophila. Polyclonal antibodies raised against triple-stranded DNA/RNA as well as the dye Thiazole Orange were used for staining of fixed tissue. Stainings were conducted with metaphase chromosome spreads from neuroblasts, spreads of polythene chromosomes from larval salivary glands. For comparison, tissue from bw-dominant larvae were stained. bwD flies contains tandom inserts at a specific position on the chromosome 2. 

In addition to the antibody staining, Dr. Gorab uses in situ hybridisation on not-denatured preparations to test the presence of unpaired DNA containing the AAGAG repeats. 

The problem with a histological approach is the specificity of the staining, both with an antibody as well as the dye. Even if the antibody has been extensively characterised to a certain degree in other experimental settings, one can in no way take it for granted that the antibody will be specific in different experimental settings. It is absolutely required to demonstrate the specificity of the antibody staining as well as the Thiazole orange staining with the chromosome spreads used in the study. If the specificity is not demonstrated, the meaning to the staining remains unclear. 

Without demonstration of specificity, the manuscript must not be published. 

Specific remarks which came across when reading and working through the paper: I did not take the effort to make this list complete. 

1. In figure 2 stainings of polytene chromosomes with the triple DNA/RNA antibody are shown. Due to the structure of these chromosome spreads, the signal should appear as a band, as seen in Figure 5 b. The signals in Fig. 2b and 2f are  however clearly circular. Why is no band visible?

2. The concentration of the antibody is not stated. A „dilution of 1:50“ is not informative and does not allow to repeat the experiments. 

3. The advantage of polytene chromosomes are employed because of their banding pattern allowing easy assignments of genetic positions. The only case where an assignment is shown is Fig. 6. The data would be more convincing if banding pattern for the all polytean chromosome spreads would be shown. The banding pattern in Fig. 6 is not shown, i. e. that the red signal locates to the 59 region. One has to rely on the author. 

Author Response

Reviewer 2

 The problem with a histological approach is the specificity of the staining, both with an antibody as well as the dye. Even if the antibody has been extensively characterised to a certain degree in other experimental settings, one can in no way take it for granted that the antibody will be specific in different experimental settings. It is absolutely required to demonstrate the specificity of the antibody staining as well as the Thiazole orange staining with the chromosome spreads used in the study. If the specificity is not demonstrated, the meaning to the staining remains unclear. 

Without demonstration of specificity, the manuscript must not be published. 

Response: I thank you for reading the manuscript as well as for your comments. This work represents an effort to improve the methodological repertoire for detecting non-canonical DNA structures in chromosomes. In this sense, I think it addresses your criticism on the lack of a better experimental design for the assessment of triplexes in vivo. If it is not yet good to your satisfaction, refinements can be made eventually on the basis of improvements described in the manuscript that could only be made thanks to published data obtained by our group. If you look at that previous report (Gorab et al., Chromosome Res. 2009), antibody specificity was first demonstrated in detail with ELISA assays. In addition, essentially the same experimental setting as exploited in this submission was used there showing results in Drosophila as well as in other dipteran. In relation to TO specificity, it was also demonstrated using the same experimental setting (Gorab & Pearson, J. Histochem. Cytochem. 2017/18). Additional biochemical data on TO specificity were also published by other groups (O'Neil et al., Electrophoresis. 2018, 39:1474; Lubitz et al., Biochemistry, 2010, 49:3567).

 Specific remarks which came across when reading and working through the paper: I did not take the effort to make this list complete. 

1. In figure 2 stainings of polytene chromosomes with the triple DNA/RNA antibody are shown. Due to the structure of these chromosome spreads, the signal should appear as a band, as seen in Figure 5 b. The signals in Fig. 2b and 2f are  however clearly circular. Why is no band visible?

Response: These are cases of very strong antibody labeling so that the typical band morphology is not seen and instead expanded signals are detected. Images in figure 2 were chosen just because there were two polytene complements in the same field showing the same labelling pattern.

 2. The concentration of the antibody is not stated. A „dilution of 1:50“ is not informative and does not allow to repeat the experiments. 

Response: Antibody concentration (0.5 mg/ml) was added in Materials and Methods of the new manuscript version.

 3. The advantage of polytene chromosomes are employed because of their banding pattern allowing easy assignments of genetic positions. The only case where an assignment is shown is Fig. 6. The data would be more convincing if banding pattern for the all polytean chromosome spreads would be shown. The banding pattern in Fig. 6 is not shown, i. e. that the red signal locates to the 59 region. One has to rely on the author. 

Response: Part of the classical drawing by Bridges was included to facilitate identification of the only chromosome section where brown dominant allele is located. For this reason just this section appears illustrated by the Bridges’ drawing as no other euchromatic region carries the AAGAG repeat block regarding such a specific mutant. The banding pattern for all polytene chromosomes could be useful provided that multiple signals were detected along them. But this is not the case.

 Reviewer 3 Report

The authors present an interesting and more convenient method to detect triplex DNA in vivo using a common organic fluorescent dye. The authors present their experiments and results very clearly.

My major concern, however, is if the authors have conducted experiments to assess a) the optimized conditions for staining with TO and b) how they authors ensure that TO binds specifically only to triplex formations.

In more details,

a) the authors present that TO staining gives weaker fluorescence signal than that of immunological detection by Abs. I observed that the authors use only 5 min incubation for staining with TO. Is that enough time to get a good signal? The authors should provide more details about the optimization conditions for TO staining, eg incubation time, different diluents, including negative controls.

b) According to other reports (O'Neil et al., Electrophoresis. 2018 Jun;39(12):1474-1477 and Lubitz et al., Biochemistry201049 (17):3567–3574) TO also  binds to double stranded DNA, but instead of other intercalating dyes, TO also binds strongly to triplex and quadruplex forms. How the authors ensure that TO binds only to triplex nucleic acids sequences? The authors should convince the readers more about that issue. The authors should also provide the negative control images with TO staining. 

Author Response

Reviewer 3

 My major concern, however, is if the authors have conducted experiments to assess a) the optimized conditions for staining with TO and b) how they authors ensure that TO binds specifically only to triplex formations.

Response: Thank you very much for reviewing the manuscript and for your comments.

 In more details,

a) the authors present that TO staining gives weaker fluorescence signal than that of immunological detection by Abs. I observed that the authors use only 5 min incubation for staining with TO. Is that enough time to get a good signal? The authors should provide more details about the optimization conditions for TO staining, eg incubation time, different diluents, including negative controls.

Response: Our previous report (J. Histochem. Cytochem. 2017-2018) describes in detail the staining properties of TO in chromosomes. In short, TO concentration and incubation time are critical for detecting specifically triplex DNA. Materials and Methods described in this submission contain the protocol for specific triplex DNA detection. TO incubations longer than 5 min will stain the whole chromosome as published previously.

 b) According to other reports (O'Neil et al., Electrophoresis. 2018 Jun;39(12):1474-1477 and Lubitz et al., Biochemistry, 2010, 49 (17):3567–3574) TO also  binds to double stranded DNA, but instead of other intercalating dyes, TO also binds strongly to triplex and quadruplex forms. How the authors ensure that TO binds only to triplex nucleic acids sequences? The authors should convince the readers more about that issue. The authors should also provide the negative control images with TO staining. 

Response: That is correct and we showed this in our previous report (J. Histochem. Cytochem. 2017-2018) except that quadruplex DNA could not be detected in human chromosome ends as expected. We took advantage of the anti-triplex antibody that does not bind to quadruplex DNA so that our TO results are related just to triplexes. As explained above, higher TO concentration and/or longer incubation time stain double-stranded DNA as DAPI does. Images contained in this manuscript provide negative controls as proper TO concentration and incubation time restrict staining to those areas that were labelled by anti-triplex antibodies. In fact, your concern was addressed carefully in a more specific article on TO properties in chromosomes that was mentioned above.

Round  2

Reviewer 1 Report

Most of the concerns I raised were nicely addressed by the author. But, there are a few points that have not responded successfully.

 Major points 1:

To make it clear the connection between AAGAG-repeats and triplex DNA in vivo, it is better to perform double staining and provide the direct evidence showing that AAGAG-repeats colocalize with the signal of the antibodies against triplex DNA and/or TO.

A new supplementary figure (Fig. S3) is not the direct evidence showing that AAGAG-repeats form triplex DNA in vivo. As the author argue in the abstract that “AAGAG tandem satellite repeats are triplex-forming sequences.”, it is important to show whether the signal of in situ hybridization against AAGAG-repeats and the signal of the antibodies against triplex DNA and/or TO can be detected at the same sites on chromosomes.

 Minor points 5:

“Ribosomal DNA probe added as a control….” in lines 176-177. It would be better to provide this control data including the results of rDNA with or without heat denaturation in Supplementary Figures.

A new supplementary figure (Fig. S6) shows the results after heat denaturation. What is really important is to show is that only AAGAG but not rDNA is detected by in situ hybridization under the same experimental condition when chromosomal DNA is not heat denatured.

 Minor points 7:

It is not easy to recognize the signal of antibodies and TO in Figure 5. It is better to make the Figure 5 brighter.

I cannot see the difference.

Author Response

Major points 1:

To make it clear the connection between AAGAG-repeats and triplex DNA in vivo, it is better to perform double staining and provide the direct evidence showing that AAGAG-repeats colocalize with the signal of the antibodies against triplex DNA and/or TO.

A new supplementary figure (Fig. S3) is not the direct evidence showing that AAGAG-repeats form triplex DNA in vivo. As the author argue in the abstract that “AAGAG tandem satellite repeats are triplex-forming sequences.”, it is important to show whether the signal of in situ hybridization against AAGAG-repeats and the signal of the antibodies against triplex DNA and/or TO can be detected at the same sites on chromosomes.

Response: I thank you for reviewing again the manuscript. The brown dominant mutation provides a direct evidence for the helical features of AAGAG-repeats since they have been described as the only repeats translocated into section 59E.  Such a new supplementary figure has exactly what you were asking me to show in your above first paragraph: the overlap of antibody and in situ hybridisation signals. Double labelling performed in mitotic chromosomes from wild type Drosophila will also show the overlap but the compaction degree of chromosomes will not warrant that the antibody targeted AAGAG repeats as there are other repeat types in centromeric/pericentric regions. By exploiting the brown dominant mutation in polytene chromosomes, there is no doubt in relation to the antibody target.

Minor points 5:

“Ribosomal DNA probe added as a control….” in lines 176-177. It would be better to provide this control data including the results of rDNA with or without heat denaturation in Supplementary Figures.

A new supplementary figure (Fig. S6) shows the results after heat denaturation. What is really important is to show is that only AAGAG but not rDNA is detected by in situ hybridization under the same experimental condition when chromosomal DNA is not heat denatured.

Response: Figure S6 was included according to your request stated in your first pararaph showing in situ hybridisation results of the two probes when chromosomal DNA undergoes denaturation (“It would be better to provide this control data”, …). Omitting denaturation but keeping the same experimental condition, only the satellite probe is detected as seen in Figures 4 and S5. FITC channel was not included in the figures as no hybridisation signal is detected.

Minor points 7:

It is not easy to recognize the signal of antibodies and TO in Figure 5. It is better to make the Figure 5 brighter.

I cannot see the difference.

Response:  Sorry for not understanding your initial observation. I think chromosomes can be better seen in this new version.

Reviewer 2 Report

The revised version of the manuscript includes a few changes within the text and corrections to specific comments of the reviewers. The main part of the manuscript including the figures has not been changed.

My central concern - over interpretation of the data has not been addressed by the revision. Photographs of histological stainings are presented that are supposed to be specific for triple helical DNA. The author argues that the antibody used has been shown in other and previous studies to be specific. In the current manuscript NO controls for the specificity of the stainings are shown. In the revised version, no controls for specificity have been added.

Such internal controls are essential to make the data conclusive and convincing. It is a common and well-known problem in immune-histology that antibodies are have limited specificity. This specificity depends very much on the assay and the experimental conditions, such as fixation, for example. An antibody can be highly specific in western blot but poorly in immunohistology. Ideally, wild type and mutant embryos lacking the antigen and staining together within the same tube are presented side-by-side. I know that is is not easily done in the case of triple helices. But some sort of convincing specificity control which has been conducted in parallel or within the same tube should be presented as part of the experimental data. A simple reference to previous publications is not convincing, since staining, especially in this case, depends on the experimental conditions. 

Figure 2 remains unconvincing due to the „dots“. I would expect bands of the polythene chromosomes. Photographs should be shown in which bands are visible.

Author Response

Reviewer 2

 The revised version of the manuscript includes a few changes within the text and corrections to specific comments of the reviewers. The main part of the manuscript including the figures has not been changed.

 My central concern - over interpretation of the data has not been addressed by the revision. Photographs of histological stainings are presented that are supposed to be specific for triple helical DNA. The author argues that the antibody used has been shown in other and previous studies to be specific. In the current manuscript NO controls for the specificity of the stainings are shown. In the revised version, no controls for specificity have been added.

Such internal controls are essential to make the data conclusive and convincing. It is a common and well-known problem in immune-histology that antibodies are have limited specificity. This specificity depends very much on the assay and the experimental conditions, such as fixation, for example. An antibody can be highly specific in western blot but poorly in immunohistology. Ideally, wild type and mutant embryos lacking the antigen and staining together within the same tube are presented side-by-side. I know that is is not easily done in the case of triple helices. But some sort of convincing specificity control which has been conducted in parallel or within the same tube should be presented as part of the experimental data. A simple reference to previous publications is not convincing, since staining, especially in this case, depends on the experimental conditions. 

Response: I thank you for reviewing again de MS as well as for your suggestion but embryo labelling has to be standardised for antibody as well as dye. This can be done but, even so, use of polytene chromosomes is essential for sequence diagnostic, otherwise the antibody target remains undetermined. The following information describes a control that was carried out in the past when we started the first immunocytochemical experiments. It ensured that the antibody dilution has only IgG reactive with triple helices and nothing else that is able to stain chromosomes. Such an additional, unpublished control for antibody specificity in situ was performed by adding either poly(dT).poly(dA).poly(dT), or poly(dT).poly(dA).poly(rU) or  even poly(rU).poly(dA).poly(rU) complexes assembled as described previously (15) to anti-triplex antibody dilutions (approximately 200 ng per slide). Such a procedure abolished fluorescence detection chromosomes. This information is described for the first time and was included in Materials and Methods of this manuscript (sub-section 2.3).

 Figure 2 remains unconvincing due to the „dots“. I would expect bands of the polythene chromosomes. Photographs should be shown in which bands are visible.

Response: Figure 2 and its corresponding caption were remade according to your request.

Reviewer 3 Report

The authors responded clearly to all reviewers' comments.

Author Response

Response: I thank you very much again for reviewing the manuscript and for your comments.